# Design, Synthesis and Biological Evaluation of Novel N-Arylpiperazines Containing a 4,5-Dihydrothiazole Ring

**DOI:** 10.3390/ph16101483

**Published:** 2023-10-18

**Authors:** Giorgia Andreozzi, Maria Rosaria Ambrosio, Elisa Magli, Giovanni Maneli, Beatrice Severino, Angela Corvino, Rosa Sparaco, Elisa Perissutti, Francesco Frecentese, Vincenzo Santagada, Anna Leśniak, Magdalena Bujalska-Zadrożny, Giuseppe Caliendo, Pietro Formisano, Ferdinando Fiorino

**Affiliations:** 1Dipartimento di Farmacia, Università di Napoli Federico II, Via D. Montesano, 49, 80131 Naples, Italy; giorgia.andreozzi@unina.it (G.A.); bseverin@unina.it (B.S.); angela.corvino@unina.it (A.C.); rosa.sparaco@unina.it (R.S.); perissut@unina.it (E.P.); frecente@unina.it (F.F.); santagad@unina.it (V.S.); caliendo@unina.it (G.C.); 2URT “Genomic of Diabetes”, Institute for Experimental Endocrinology and Oncology “G. Salvatore”, National Research Council (IEOS-CNR), Via Pansini 5, 80131 Naples, Italy; mariarosaria.ambrosio@gmail.com (M.R.A.); pietro.formisano@unina.it (P.F.); 3Dipartimento di Sanità Pubblica, Università di Napoli Federico II, Via Pansini, 5, 80131, Naples, Italy; elisa.magli@unina.it; 4Department of Translational Medicine, University of Naples “Federico II”, Via Pansini 5, 80131 Naples, Italy; dr.giovanniromanelli@libero.it; 5Department of Pharmacotherapy and Pharmaceutical Care, Centre for Preclinical Research and Technology, Faculty of Pharmacy, Medical University of Warsaw, 1 Banacha Str., 02-097 Warsaw, Poland; anna.lesniak@wum.edu.pl (A.L.); magdalena.bujalska@wum.edu.pl (M.B.-Z.)

**Keywords:** arylpiperazines, 5-HT_1A_ ligands, binding assays, in vitro assay, cytotoxic activity

## Abstract

Arylpiperazines represent one of the most important classes of 5-HT_1A_R ligands and have attracted considerable interests for their versatile properties in chemistry and pharmacology, leading to the research of new derivatives that has been focused on the modification of one or more portions of such pharmacophore. An efficient protocol for the synthesis of novel thiazolinylphenyl-piperazines (**2a**–**c**) and the corresponding acetylated derivatives was used (**3a**–**c**). The new compounds were tested for their functional activity and affinity at 5-HT_1A_ receptors, showing an interesting affinity profile with a Ki value of 412 nM for compound **2b**. The cytotoxic activity of novel thiazolinylphenyl-piperazines (**2a**–**c**) and corresponding N-acetyl derivatives (**3a**–**c**) against human prostate and breast cancer cell lines (LNCAP, DU-145 and PC-3, MCF-7, SKBR-3 and MDA-MB231) was investigated according to the procedure described in the literature. The reported data showed a cytotoxic effect for **2a**–**c** and **3a**–**c** compounds (IC_50_ values ranging from 15 µM to 73 µM) on the investigated cancer cell lines, with no effect on noncancer cells. Future studies will be aimed to investigate the mechanism of action and therapeutic prospects of these new scaffolds.

## 1. Introduction

N-Arylpiperazines are an important class of organic compounds that have attracted considerable interest owing to their versatile properties in chemistry and pharmacology, leading to a wide development in the field of medicinal chemistry [1,2]. These scaffolds are known to possess antihistaminic, antihypertensive, adrenolytic and antiproliferative properties [1,3,4,5]. Arylpiperazine derivatives belong to one of the most important classes of molecules approved for the management of neurological disorders and act with high affinity toward serotoninergic receptors. Mechanistically, almost all of them act as agonists (partial/full) of 5-HT_1A_R, and two main interactions proved to be important for the affinity of arylpiperazines toward 5-HT_1A_Rs: (a) an ionic bond between the protonated nitrogen atom of the piperazine ring and the carboxyl oxygen of the side chain of Asp3.32 and (b) an edge-to-face CH/π interaction between the aromatic ring and the Phe6.52 residue, which stabilizes the ligand binding [6]. The basic pharmacophore of the 5-HT_1A_R ligands is the same for agonists and antagonists and consists of an aromatic nucleus and a basic nitrogen atom, whose optimal distance is 5.2 Å, while the nitrogen lies at 0.2 Å above the plane defined by the reference ring [7]. Moreover, arylpiperazine derivatives have been described to reduce prostate cancer cell growth [8] and to ameliorate sensitivity to Tamoxifen in ER+ BC cells [4]. 5-HT1AR targeting has already been described to display antitumor activity in prostate cancer cells [9]. In our laboratories, there has been ongoing research dedicated to developing more selective arylpiperazine derivatives as serotoninergic ligands [10,11,12,13,14,15,16,17,18,19] to provide novel pharmacological tools that could improve our knowledge on signal transduction mechanisms and increase affinity and selectivity.

Thus, we have designed a new arylpiperazine scaffold supporting a 4,5-dihydrothiazole substituent on all the possible positions of the aromatic ring that has never been described earlier. This choice was made to investigate how the introduction of dihydrothiazole moiety could affect the affinity/activity profile towards serotoninergic receptors, but also to verify the activities of these compounds against cancer cells. It was already reported that the thiazole nucleus showed a wide range of pharmacological activities like anti-inflammatory, anti-tubercular, anti-diabetic, anti-malarial and anti-cancer [20,21]. Consequently, it represents a perfect candidate to explore lead compounds and other drug-like molecules for a variety of disorders [20,21].

## 2. Results

In the present research, we report the synthesis of 1-(2-thiazolinylphenyl)piperazine, 1-(3-thiazolinylphenyl)piperazine and 1-(4-thiazolinylphenyl)piperazine. The synthesized compounds were preliminarily tested for their affinity to 5-HT_1A_R and evaluated together with the corresponding N-acetyl derivatives for their cytotoxic activities against human prostate cancer and breast cancer cell lines. The general strategy for the synthesis of the target compounds is summarized in Figure 1. Treatment of commercially available 4-substituted piperazines (**1a**–**c**), with 2-aminoethane-1-thiol hydrochloride in the presence of NaOH under solvent-free conditions heating to 80 °C, gave the corresponding thiazolinylphenyl-piperazines (**2a**–**c**). Subsequent treatment with acetic anhydride in diethyl ether provided the corresponding acetylated derivatives (**3a**–**c**). All the synthetized compounds were characterized by ^1^H NMR, ^13^C NMR and MS, providing data consistent with the proposed structures.

The new thiazolinylphenyl-piperazines (**2a**–**c**) were tested for their affinity with 5-HT_1A_ receptors. The obtained results (Table 1) indicated an interesting affinity profile of compound **2b** with a Ki value of 0.412 µM, while compound **2a** showed a weaker 5-HT_1A_ affinity of 2.29 µM. Additionally, a considerable reduction in 5-HT_1A_ binding was obtained when the dihydrothiazole moiety was moved to the para position as in compound **2c** (Ki = 49.5 µM). This observation underlines the relevance of placing a dihydrothiazole moiety in the meta position of the phenyl ring to improve 5-HT_1A_ receptor binding.

These data are extremely encouraging considering that they are obtained by testing the only thiazolinylphenyl-piperazine scaffold not included in the structure of long-chain arylpiperazines (LCAPs) derivatives. Therefore, this scaffold could represent a new structural element that is useful for discovering novel pharmacological tools in treatment to target 5HTR [20,21,22]. Simultaneously, it was already reported that thiazole-containing compounds have been developed as possible inhibitors of several targets involved in biochemical and oncogenic regulatory pathways, including enzyme-linked receptors located on the cell membrane (polymerase inhibitors) and cell cycle (microtubular inhibitors) [23,24]. Hence, we investigated the cytotoxic activity of novel thiazolinylphenyl-piperazines (**2a**–**c** and **3a-c**) against LNCAP (androgen sensitive), DU145 and PC3 (androgen independent) prostate cancer cells, and MCF7 (ER+, PR+, HER2−), SKBR3 (ER−, PR+, HER2+) and MDA-MB231 (ER−, PR−, HER2−) breast cancer cell lines (Table 2).

We observed that treatment with compounds **2a**–**c** resulted only in slight or no reduction in cell viability in LNCAP and PC-3 cells, with the only exception of **2c** showing an IC_50_ value of 32 µM on LNCAP, although data did not reach statistical significance. Compounds **2a**–**c** were found to be slightly more active against DU145 cells (Figure 1), with IC_50_ values ranging from 48 µM to 67 µM (Table 2).

In contrast with prostate cancer cells, all thiazolinylphenyl-piperazine compounds significantly reduce cell viability in breast cancer cell lines, with about a 50% reduction obtained with at least 25 µM concentration (Figure 2). In fact, compounds **2a**–**c** showed reasonable antitumor activity against MCF-7, SKBR-3 and MDA-MB231 breast cancer cell lines, with IC_50_ values ranging from 15 to 40 µM. In particular, the most interesting results were obtained for the thiazolinylphenyl-piperazines (**2a**–**c**) on the MCF-7 cell line, exhibiting IC_50_ values of 15, 16 and 19 µM, respectively (Table 2). Notably, a cytotoxic effect was observed on highly aggressive MDA-MB231 (with IC_50_ ranging from 31 µM to 40 µM).

In order to verify if the antitumor effects could be influenced by cell-permeability of these derivatives and to improve lipophilicity, we decided to analyze the cytotoxic activity of the corresponding N-acetyl derivatives (**3a**–**c**) on PC-3 cell line [25]. The choice was made considering that the treatment with compounds **2a**–**c** determine only a slight or no reduction in cell viability in these cells (Table 2 and Figure 1). Consequently, as shown in Figure 3, treatment with compounds **3a**–**c** significantly reduced PC-3 cell viability (IC_50_ values ranging from 32 to 73 µM). We also observed that, concerning the activity against MDA-MB231 cells, the acetates, although considered to possess higher lipophilicity, have comparable activity to compounds **2a**–**c**. These data indicated, consistently with previous observations [26], that at least in PC-3 cells, increased permeability could be due to improved lipophilicity.

Finally, in order to examine the selective toxicity of thiazolinylphenyl-piperazines (**2a**–**c**) and the corresponding N-acetyl derivatives (**3a**–**c**) on cancer cells, we verified that these compounds did not affect cell viability of non-cancer prostate (PNT-1) and breast epithelial cells (MCF-10) (Figure 4).

## 3. Discussion

This study reported the synthesis, the binding assays on 5-HT_1A_ receptors and the biological evaluation against prostate and breast cancer cell lines of a novel class of thiazolinylphenyl-piperazines that has never been described before. The compounds **2a**–**c** demonstrated sub- or micromolar 5-HT_1A_ receptor affinities dependent on the placement of the dihydrothiazole moiety in the phenyl ring, with the meta position being the most favorable. These positive results, on one hand, could serve as a valuable tool for further research on arylpiperazine derivatives displaying a high affinity/selectivity profile towards serotoninergic receptors (5-HT_1A_, 5-HT_2A_ and 5-HT_2C_) [27]. On the other hand, considering that 5-HT acts as a mitogenic and anti-apoptotic factor for a wide range of normal and tumor cells, this class of compounds may serve as a novel and valuable anti-cancer scaffold for developing a novel therapeutic approach. In fact, in the present study, our data have indicated that these compounds are able to significantly inhibit the growth of prostate and breast tumor cell lines, simultaneously showing a significant selectivity towards non-transformed cells. Therefore, given the absence of therapeutic approaches devoid of cytotoxic effects, these results highlight the potential innovation that these compounds can represent. Notably, a significant effect has been observed on androgen-independent prostate cancer and triple negative breast cancer cells, which represent models of extremely aggressive forms of tumors. Consequently, future studies will be aimed to identify their mechanism of action, and further research involving other classes of arylpiperazine derivatives as well as the preparation of long-chain arylpiperazines (LCAPs) derivatives characterized by this new scaffold are in progress.

## 4. Materials and Methods

### 4.1. Synthesis

#### 4.1.1. General Procedures

All reagents and substituted piperazines were commercial products purchased from Aldrich. Melting points were determined using a Kofler hot-stage apparatus and are uncorrected. ^1^H-NMR and ^13^C-NMR spectra (reported in the Appendix A) were recorded on a Varian Mercury Plus 400 MHz instrument. Unless otherwise stated, all spectra were recorded in CDCl3. Chemical shifts are reported in ppm using Me4Si as the internal standard. The following abbreviations are used to describe peak patterns when appropriate: s (singlet), d (doublet), t (triplet), m (multiplet), q (quartet), qt (quintet), dd (double doublet), ddd (double dd) and bs (broad singlet). Mass spectra of the final products were performed on an API 2000 Applied Biosystem mass spectrometer. Where analyses are indicated only by the symbols of the elements, results obtained are within ±0.4% of the theoretical values. All reactions were followed by TLC, carried out on Merck silica gel 60 F254 plates with a fluorescent indicator, and the plates were visualized with UV light (254 nm). Preparative chromatographic purifications were performed using silica gel column (Kieselgel 60). Solutions were dried over Na_2_SO_4_ and concentrated with a Buchi rotary evaporator at low pressure.

#### 4.1.2. General Procedure for the Synthesis of 2-(x-(Piperazin-1-yl) phenyl)-4,5-dihydrothiazole (**2a**–**c**)

In a 100 mL round-bottomed flask, a mixture of appropriate x-(1-piperazinyl) benzonitrile (**1a**–**c**, 1 mmol), 2-aminoethane-1-thiol hydrochloride (1,5 mmol) and sodium hydroxide (0,2 mmol) was stirred at 80 °C for 4 h under solvent-free conditions. Subsequently, the reaction mixtures were dissolved in dichloromethane (20 mL) and washed with water and brine. The organic layers were dried over anhydrous Na_2_SO_4_ and the solvent removed under vacuum. The crude products were purified by silica gel open chromatography using dichloromethane/methanol (8:2 *v*/*v*) as an eluent. The combined and evaporated product fractions were crystallized from die–thyl ether, yielding the desired products as white solids.

*2-(2-(Piperazin-1-yl) phenyl)-4,5-dihydrothiazole* (**2a**): Yield: 58%; mp: 78–80 °C; ^1^H-NMR (400 MHz, CDCl_3_) δ: 1.83 (s, 1H), 2.95 (bs, 4H, 2CH_2_ pip.), 3.13 (bs, 4H, 2CH_2_ pip.), 3.24(t, 2H, -CH_2_-, *J* = 8.3 Hz), 4.30 (t, 2H, -CH_2_-, *J* = 8.3 Hz), 7.10 (t, 1H, *J* = 7.7 Hz), 7.14 (d, 1H, *J* = 7.7 Hz), 7.40 (t, 1H, *J* = 7.7 Hz), 7.79 (d, 1H, *J* = 7.7 Hz); ^13^C-NMR (101 MHz, CDCl_3_) δ: 33.28, 45.80, 54.35, 62.83, 119.77, 123.34, 129.39, 130.04, 131.24, 152.07, 168.17 ESI-MS *m*/*z* [M + H]^+^ calculated for C_13_H_17_N_3_S 247.36, Found = 248.1 Anal. Calcd for C_13_H_17_N_3_S: C, 63.12; H, 6.93; N, 16.99. Found C, 63.30; H, 6.95; N, 17.04.*2-(3-(Piperazin-1-yl) phenyl)-4,5-dihydrothiazole* (**2b**): Yield: 65%; mp: 109–110 °C; ^1^H-NMR (400 MHz, CDCl_3_) δ: 1.80 (s, 1H), 3.06 (m, 4H, 2CH_2_ pip.), 3.22 (m, 4H, 2CH_2_ pip.), 3.41 (t, 2H, -CH_2_-, *J* = 8.4 Hz), 4.46 (t, 2H, -CH_2_-, *J* = 8.4 Hz), 7.15 (bs, 1H), 7.29 (bs, 2H), 7.45 (bs, 1H); ^13^C-NMR (101 MHz, CDCl_3_) δ: 33.60, 46.07, 50.18, 65.18, 115.15, 118.86, 120.07, 129.20, 134.03, 151.79, 169.00 ESI-MS *m*/*z* [M + H]^+^ calculated for C_13_H_17_N_3_S 247.36, Found = 248.14 Anal. Calcd. for C_13_H_17_N_3_S: C, 63.12; H, 6.93; N, 16.99. Found C, 63.24; H, 6.73; N, 17.05.*2-(4-(Piperazin-1-yl) phenyl)-4,5-dihydrothiazole* (**2c**): Yield: 60%; mp: 152–153 °C; ^1^H-NMR (400 MHz, CDCl_3_) δ: 1.80 (s, 1H), 3.02 (m, 4H, 2CH_2_ pip.), 3.26 (m, 4H, 2CH_2_ pip.), 3.36 (t, 2H, -CH_2_-, *J* = 8.2 Hz), 4.42 (t, 2H, -CH_2_-, *J* = 8.2 Hz), 6.88 (d, 2H, *J* = 8.4 Hz), 7.24 (d, 2H, *J* = 8.4 Hz); ^13^C-NMR (101 MHz, CDCl_3_) δ: 33.53, 45.88, 49.04, 64.97, 114.39, 123.86, 129.70, 153.44, 167.32 ESI-MS *m*/*z* [M + H]^+^ calculated for C_13_H_17_N_3_S: 247.36, Found = 248.13 Anal. Calcd. for C_13_H_17_N_3_S: C, 63.12; H, 6.93; N, 16.99. Found C, 62.99; H, 6.95; N, 16.97.

#### 4.1.3. General Procedure for the Synthesis of 1-(4-(x-(4,5-Dihydrothiazol-2-yl) phenyl)piperazin-1-yl)ethan-1-one (**3a**–**c**):

In a 100 mL round-bottomed flask, a mixture of appropriate 2-(x-(piperazin-1-yl)phenyl)-4,5-dihydrothiazole (**2a**–**c**) (1 mmol) and acetic anhydride (1, 2 mmol) was stirred at room temperature for two hours. After completion, the reaction mixtures were dissolved in diethyl ether (20 mL) and washed with water and brine. The organic layers were dried over anhydrous Na_2_SO_4_ and the solvent removed under vacuum. The crude products were purified by silica gel open chromatography using dichloromethane/methanol (9.5:0.5 *v*/*v*) as an eluent. The combined and evaporated product fractions were crystallized from diethyl ether, yielding the desired products as white solids.

*1-(4-(2-(4,5-Dihydrothiazol-2-yl) phenyl)piperazin-1-yl)ethan-1-one* (**3a**): Yield: 62%; mp: 90–92 °C; ^1^H-NMR (400 MHz, CDCl_3_) δ: 2.15 (s, 3H), 2.98 (bs, 4H, 2CH_2_ pip.), 3.26 (t, 2H, -CH_2_-, *J* = 8.0 Hz) 3.72 (bs, 2H, -CH_2_ pip.), 3.88 (bs, 2H, -CH_2_ pip.), 4.32 (t, 2H, -CH_2_-, *J* = 8.0 Hz), 7.10 (d, 1H, *J* = 7.8 Hz), 7.15 (t, 1H, *J* = 7.8 Hz), 7.41 (t, 1H, *J* = 7.8 Hz), 7.82 (d, 1H, *J* = 7.8 Hz); ^13^C-NMR (101 MHz, CDCl_3_) δ: 21.40, 33.30, 41.35, 46.26, 52.58, 53.39, 63.01, 119.89, 124.01, 129.55, 130.26, 131.34, 150.94, 167.64, 169.21 ESI-MS *m*/*z* [M + H]^+^ calculated for C_15_H_19_N_3_OS 289,40 Found = 290.2 Anal. Calcd. for C_15_H_19_N_3_OS: C, 62.26; H, 6.62; N, 14.52. Found C, 62.44; H, 6.59; N, 14.56.*1-(4-(3-(4,5-Dihydrothiazol-2-yl)phenyl)piperazin-1-yl)ethan-1-one* (**3b**): Yield: 65%; mp: 91–93 °C; ^1^H-NMR (400 MHz, CDCl_3_) δ: 2.16 (s, 3H), 3.22 (t, 2H, CH_2_ pip., *J* = 5.2 Hz), 3.25 (t, 2H, CH_2_ pip., *J* = 5.2 Hz), 3.42(t, 2H, -CH_2_-, *J* = 8.4 Hz) 3.64 (t, 2H, -CH_2_ pip., *J* = 5.2 Hz), 3.79 (t, 2H, -CH_2_ pip. *J* = 5.2 Hz), 4.46 (t, 2H, -CH_2_-, *J* = 8.4 Hz), 7.04 (m, 2H), 7.32 (bs, 1H,), 7.46 (bs, 1H); ^13^C-NMR (101 MHz, CDCl_3_) δ: 21.37, 33.66, 41.29, 46.14, 49.20, 49.41, 65.17, 115.65, 119.35, 120.90, 129.38, 134.19, 150.95, 168.92, 169.19 ESI-MS *m*/*z* [M + H]^+^ calculated for C_15_H_19_N_3_OS 289,40 Found = 290.2 Anal. Calcd. for C_15_H_19_N_3_OS: C, 62.26; H, 6.62; N, 14.52. Found C, 62.07; H, 6.63; N, 14.50.*1-(4-(4-(4,5-Dihydrothiazol-2-yl)phenyl)piperazin-1-yl)ethan-1-one* (**3c**): Yield: 61%; mp: 199–200 °C; ^1^H-NMR (400 MHz, CDCl_3_) δ: 2.16 (s, 3H), 3.29 (bs, 2H, CH_2_ pip.), 3.33 (bs, 2H, CH_2_ pip.), 3.39 (t, 2H, -CH_2_-, *J* = 8.2 Hz) 3.65 (bs, 2H, -CH_2_ pip.), 3.80 (bs, 2H, -CH_2_ pip.), 4.42 (t, 2H, -CH_2_-, *J* = 8.2 Hz), 6.90 (d, 2H, *J* = 8.2 Hz), 7.77 (d, 2H, *J* = 8.2 Hz); ^13^C-NMR (101 MHz, CDCl_3_) δ: 21.33, 33.60, 41.06, 45.92, 48.03, 48.32, 65.03, 114.86, 124.69, 129.78, 152.58, 167.67, 169.26 ESI-MS *m*/*z* [M + H]^+^ calculated for C_15_H_19_N_3_OS 289,40 Found = 290.2 Anal. Calcd. for C_15_H_19_N_3_OS: C, 62.26; H, 6.62; N, 14.52. Found C, 62.50; H, 6.60; N, 14.55.

### 4.2. In Vitro Receptor Binding

#### 4.2.1. Membrane Preparation

Sprague Dawley rats were sacrificed by isoflurane overdose. Brains were rapidly removed and placed on ice. Hippocampi (for 5-HT_1A_ assays) and frontal cortices (for 5-HT_2A_ assays) were dissected on a Petri dish. The tissue from 10 rats was homogenized in 30 vol. homogenization buffer (50 mM Tris-HCl, pH = 4.7, 1mM EDTA, 1mM dithiothreitol) with a handheld teflon-glass homogenizer. The homogenate was centrifuged at 48,000× *g* at 4 °C for 15 min. The pellet was suspended and homogenized in homogenization buffer and incubated for 10 min. at 36 °C. The centrifugation and suspension steps were repeated twice. The final pellet was homogenized in a 5 vol., 50 mM Tris-HCl, pH = 7.4 buffer and stored at -80 ^°^C for no longer than 6 months.

#### 4.2.2. Competitive 5-HT_1A_ Assay

For the 5-HT1A assay, ten concentrations equally spaced on a log scale (10^−14^ M–10^−5^ M) of each compound were incubated in duplicate with 1 nM [3H]8-OH-DPAT (specific activity: 200 Ci/mmol, Perkin Elmer, MA, USA) for 60 min. at 36 °C in a 50 mM Tris-HCl (pH 7.4) buffer, supplemented with 0.1% ascorbate, 5 mM MgCl_2_ and 80 µg of hippocampal membrane suspension. Non-specific binding was determined with 10 μM serotonin. The final DMSO concentration in the assay was 5%. After incubation, the reaction mixture was deposited with the FilterMate-96 Harvester (Perkin Elmer, MA, USA) onto Unifilter^®^ GF/C plates (Perkin Elmer, MA, USA) presoaked in 0.4% PEI for 1 h. Each well was washed with 2 mL of 50 mM Tris-HCl (pH 7.4) buffer to separate bound ligands from free ones. Plates were left to dry overnight. Then, 35 µL of Microscint-20 scintillation fluid (Perkin Elmer, MA, USA) was added to each filter well and left to equilibrate for 2 h. Filter-bound radioactivity was counted in a MicroBeta2 LumiJet scintillation counter (Perkin Elmer, MA, USA). Binding curves were fitted with one site non-linear regression. Binding affinity (pKi and Ki) for each compound was calculated from the EC_50_ values with the Cheng–Prusoff equation from two separate experiments.

### 4.3. Cytotoxic Activity

Media, serum and antibiotics for cell culture were from Lonza (Basel, Switzerland). MCF7 (ER+, PR+, HER2−), SKBR3 (ER−, PR+, HER2+) and MDA-MB231 (ER−, PR−, HER2−) human breast cancer cells; MCF10A non-cancer breast epithelial cells; LNCAP (androgen sensitive), DU145 and PC3 (androgen independent) human prostate cancer cells; and PNT1 non-cancer prostate epithelial cells were available in our laboratory. MCF7, SKBR3 and MDA-MB231 cells were cultured in DMEM, supplemented with 10% FBS, 2 mM glutamine, 100 units/mL penicillin and 100 units/mL streptomycin. MCF10A cells were cultured in MEBM and supplemented with 0.4% BPE, 0.1% hEGF, 0.1%, Insulin, 0.1% Hydrocortisone and 0.1% GA-1000. LNCAP and DU145 cells were cultured in RPMI and supplemented with 10% FBS, 2 mM glutamine, 100 units/mL penicillin and 100 units/mL streptomycin. PC3 and PNT1 cells were cultured in DMEM-F12 (1:1), supplemented with 10% FBS, 2 mM glutamine, 100 units/mL penicillin and 100 units/mL streptomycin. Cultures were maintained in a humidified atmosphere of 95% air and 5% CO_2_ at 37 °C. Treatment with compounds were carried out in culture conditions. For cell survival assay, cells were fixed with 50% trichloroacetic acid for at least 2 h at 4 °C, washed with distilled and de-ionized water, air-dried and stained for 30 min with 0.4% sulforhodamine B in 1% acetic acid. Unbound dye was removed, and 10 mM tris-HCl solution (pH 7.5) was added to dissolve the protein-bound dye. Cell survival was assessed by optical density determination at 510 nm using a microplate reader [28].

## Data Availability

Data is contained within the article.

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
