# Peer review of "Design, Synthesis and Biological Evaluation of Novel N-Arylpiperazines Containing a 4,5-Dihydrothiazole Ring"

_pharmaceuticals, 2023, doi:10.3390/ph16101483_

Round 1

Reviewer 1 Report

pharmaceuticals-2635195

This manuscript describes the preparation of three new arylpiperazines and their corresponding acetates and their biological evaluation concerning their affinity for 5-HT1A receptor as well as against a panel of cancer (and non-cancer) cell lines.

The authors focus (even from the first line of the abstract) in the study of arylpiperazines as important 5-HT ligands. They study the affinity of the three new compounds for the 5-HT1A receptor, but then, they present a more extensive study of these derivatives as antiproliferative agents, against prostate and breast cancer cell lines.

Is there a rationale for the study of compounds with 5-HT1AR affinity as cytotoxic agents? If not, I do not think that both biological activities should appear in the same manuscript and since in the present manuscript the study of the activity on the 5-HT receptor is not adequately developed, I would propose to omit this section and focus on the antiproliferative activity of the new compounds. Otherwise a strong evidence for the relation of the two activities should appear in the introduction and more importantly in the experimental and the results and discussion sections.

Overall, the chemical methods are not innovative, but the results of the antiproliferative activity may be of interest in the scientific community. Please find below some suggestions and comments to be addressed.

Page 1, line 32:  considerable interest

Page 1, line 36:  and act with

Page 1, line 42: A figure showing the interactions of the piperazine and aryl moieties with the receptor could be inserted in this section.

Page2 , line 52: dihydrothiazole substituent on all the possible positions of the aromatic ring

Page 2, line 61: we report the synthesis

Page 2, line 68: in the presence

Page 2, line 72: providing data

Page 3, lines 85-89: this part needs to be rephrased for clarity.

Page 3, line 100: treatment with compounds 2a-c resulted only in slight

Page 3, line 103: since the activity against DU145 cells is not in fact “significant”, I would propose to rephrase as: “compounds 2a-c were found slightly more active against DU145 cells (Figure 1) with…”

Page 4, line 112: In contrast with prostate cancer cells (same remark in Page 5, line 134).

Page 5, line 131: instead of “determine only a slight or no reduction”, better use “practically showed no reduction”.

There is a question that should be commented by the authors concerning the whole paragraph in page 5. Why the authors have chosen to study the acetates on the PC3 cell line where no activity was detected from the free bases and not the other two cell lines as would be reasonable? Again in this case, the activity of 3a-c cannot be considered “significant”. We also observe that, concerning the activity against MDA-MB2312 cells, the acetates (considered to possess higher lipophilicity) have comparable activity to compound 2a-c.

Page 5, line 121: on BCa cell viability

Page 7, lines 159-163: This comment (on the spatial orientation of the substituent) needs to be further explored and the experiments reported in the manuscript do not fully support it. Molecular modeling experiments could be helpful.

Page 7, line 197:  at 80 °C

Page 7, lines 206-217 and page 8 lines 218-223: The experimental details for the preparation of 2b and 2c are identical to those used for the preparation of 2a and should not be repeated. The same should apply for compounds 3b and 3c.

The English language needs only minor editing.

Author Response

We thank the referee for all the valuable suggestions to which we have responded as follows:

Is there a rationale for the study of compounds with 5-HT1AR affinity as cytotoxic agents? If not, I do not think that both biological activities should appear in the same manuscript and since in the present manuscript the study of the activity on the 5-HT receptor is not adequately developed, I would propose to omit this section and focus on the antiproliferative activity of the new compounds. Otherwise a strong evidence for the relation of the two activities should appear in the introduction and more importantly in the experimental and the results and discussion sections.

We followed the referee's recommendation by adding a sentence and the bibliography in the text

Page 1, line 32:  considerable interest We have modified according to the referee's indication

Page 1, line 36:  and act with We have modified according to the referee's indication

Page2 , line 52: dihydrothiazole substituent on all the possible positions of the aromatic ring

We have modified according to the referee's indication

Page 2, line 61: we report the synthesis We have modified according to the referee's indication

Page 2, line 68: in the presence We have modified according to the referee's indication

Page 2, line 72: providing data We have modified according to the referee's indication

Page 3, lines 85-89: this part needs to be rephrased for clarity. We have modified according to the referee's indication

Page 3, line 100: treatment with compounds 2a-c resulted only in slight We have modified according to the referee's indication

Page 3, line 103: since the activity against DU145 cells is not in fact “significant”, I would propose to rephrase as: “compounds 2a-c were found slightly more active against DU145 cells (Figure 1) with…” We have modified according to the referee's indication

Page 4, line 112: In contrast with prostate cancer cells (same remark in Page 5, line 134). We have modified according to the referee's indication

Page 5, line 131: instead of “determine only a slight or no reduction”, better use “practically showed no reduction”. We have modified according to the referee's indication

There is a question that should be commented by the authors concerning the whole paragraph in page 5. Why the authors have chosen to study the acetates on the PC3 cell line where no activity was detected from the free bases and not the other two cell lines as would be reasonable? Again in this case, the activity of 3a-c cannot be considered “significant”. We also observe that, concerning the activity against MDA-MB2312 cells, the acetates (considered to possess higher lipophilicity) have comparable activity to compound 2a-c. We have modified according to the referee's indication

Page 5, line 121: on BCa cell viability We have modified according to the referee's indication

Page 7, line 197:  at 80 °C We have modified according to the referee's indication

Page 7, lines 206-217 and page 8 lines 218-223: The experimental details for the preparation of 2b and 2c are identical to those used for the preparation of 2a and should not be repeated. The same should apply for compounds 3b and 3c. We have modified according to the referee's indication

Concerning the molecular modelling experiments We thank the referee, but these studies will be developed and included in an extensive work that includes several derivatives.

Reviewer 2 Report

- in the title please correct (dihydrothiazol)

- The style and font of scheme 1 need to be adjusted.

-Tables 1 and 2 need to be centered.

- The reference is missing in the anticancer screening.

-The tested compounds showed weak anticancer activities and no need to add extra discussion about that. Figures1-4 should be transferred to supplementary data.

- The structures and integration need to be added to the NMR charts in supp file.

Sentences need to be carefully revised and corrected for typo and grammar errors.

Author Response

We thank the referee for all the valuable suggestions to which we have responded as follows:

- in the title please correct (dihydrothiazol) We have modified according to the referee's indication

- The style and font of scheme 1 need to be adjusted. We have modified according to the referee's indication

-Tables 1 and 2 need to be centered. We have modified according to the referee's indication

- The reference is missing in the anticancer screening. The reference was added according to the referee's indication

- The structures and integration need to be added to the NMR charts in supp file. We have modified according to the referee's indication

Reviewer 3 Report

The manuscript entitled “Design, synthesis, and biological evaluation of novel N-arylpiperazines containing a 4,5 dihydrotiazole ring” by G. Andreozzi et al. described the synthesis, the assays on 5-HT1A receptors and the anticancer evaluation against prostate and breast cancer cell lines of a novel thiazolinylphenylpiperazines. The manuscript may be of general interest to the researchers of this field, but the manuscript lacks some information that the authors should consider and incorporate in the present form of the manuscript.

1.    The manuscript’s introduction is laconic. Only 3 of 20 references are over the past five years. The following question arises: is the topic chosen by the authors in the manuscript relevant? The authors should prove this with relevant new references for 2019-2023.

2.     The discussion is presented as conclusions and the conclusions are presented as a discussion.

3.   Scheme 1 is of low quality. It should be redesigned as follows for better readability. Please provide the general scheme of the synthesis. The table below should be presented: reagent 1, reagent 2, product.  Reaction conditions should be given below the table.

4. Supplementary materials must be mentioned in the main text of the manuscript.  The citation format of Supplementary should be in accordance with the requirements of the journal.  The title of the manuscript, authors, and affiliations should be added at the beginning of the Supplementary readability in the future. The proton signals in the 1H NMR spectra of all compounds in the Supplementary should be integrated.

5. What do pKi, Ki and CI mean? The units of measurement for these parameters should be indicated in Table 1 and the values in the text and table should be checked for agreement. What about pKi, Ki results for the compounds 3a-c?

6.  Items 4.1.2 to 4.2.6 should be reorganised. Repetition of the method of preparation for each compound is not necessary. The general procedure for the synthesis of compounds 2a-c and 3a-c should be only presented. After each of them, the characteristics of the corresponding compounds should be given. The names of the compounds should be capitalised.

7.  “CDCl3” should be “CDCl3”, “13C-NMR” should be “13C NMR”, “1H-NMR” should be “1H NMR”, “CH2” should be “CH2” etc.

Author Response

We thank the referee for all the valuable suggestions to which we responded as follows:

  1.    The manuscript’s introduction is laconic. Only 3 of 20 references are over the past five years. The following question arises: is the topic chosen by the authors in the manuscript relevant? The authors should prove this with relevant new references for 2019-2023. We have modified according to the referee's indication
  2. The discussion is presented as conclusions and the conclusions are presented as a discussion. We have modified according to the referee's indication

  1.   Scheme 1 is of low quality. It should be redesigned as follows for better readability. Please provide the general scheme of the synthesis. The table below should be presented: reagent 1, reagent 2, product. Reaction conditions should be given below the table. We have modified according to the referee's indication

  1. Supplementary materials must be mentioned in the main text of the manuscript. The citation format of Supplementary should be in accordance with the requirements of the journal. The title of the manuscript, authors, and affiliations should be added at the beginning of the Supplementary readability in the future. The proton signals in the 1H NMR spectra of all compounds in the Supplementary should be integrated. We have modified according to the referee's indication
  2. What do pKi, Ki and CI mean? The for these parameters should be indicated in Table 1 and the values in the text and table should be checked for agreement. What about pKi, Ki results for the compounds 3a-c? We have modified the units of measurement in the table according to the referee's indication
  3.  Items 4.1.2 to 4.2.6 should be reorganised. Repetition of the method of preparation for each compound is not necessary. The general procedure for the synthesis of compounds 2a-c and 3a-c should be only presented. After each of them, the characteristics of the corresponding compounds should be given. The names of the compounds should be capitalised. We have modified according to the referee's indication
  4.  “CDCl3” should be “CDCl3”, “13C-NMR” should be “13C NMR”, “1H-NMR” should be “1H NMR”, “CH2” should be “CH2” etc. We have modified according to the referee's indication

Round 2

Reviewer 1 Report

The manuscript has been substantially improved and could be accepted for publication in Pharmaceuticals.

Only minor editing is needed.

Reviewer 2 Report

The manuscript is appropriate for publication 

Reviewer 3 Report

The authors significantly improved the quality of the paper according to reviewer's concerns. I suggest the paper can be accepted.